# Microglia Reduce Herpes Simplex Virus 1 Lethality of Mice with Decreased T Cell and Interferon Responses in Brains

**DOI:** 10.3390/ijms222212457

**Published:** 2021-11-18

**Authors:** Meng-Shan Tsai, Li-Chiu Wang, Hsien-Yang Tsai, Yu-Jheng Lin, Hua-Lin Wu, Shun-Fen Tzeng, Sheng-Min Hsu, Shun-Hua Chen

**Affiliations:** 1Institute of Basic Medical Sciences, College of Medicine, National Cheng Kung University, Tainan 701, Taiwan; kihkk245@gmail.com (M.-S.T.); halnwu@ncku.edu.tw (H.-L.W.); 2School of Medicine, I-Shou University, Kaohsiung 824, Taiwan; statolish@isu.edu.tw; 3Department of Ophthalmology, Tzu Chi Hospital, Taichung 427, Taiwan; tc1512901@tzuchi.com.tw; 4Department of Microbiology and Immunology, College of Medicine, National Cheng Kung University, Tainan 701, Taiwan; ca130705@gmail.com; 5Department of Biochemistry and Molecular Biology, College of Medicine, National Cheng Kung University, Tainan 701, Taiwan; 6Department of Life Sciences, College of Biological Science and Biotechnology, National Cheng Kung University, Tainan 701, Taiwan; stzeng@ncku.edu.tw; 7Department of Ophthalmology, College of Medicine, National Cheng Kung University, Tainan 701, Taiwan

**Keywords:** herpes simplex virus 1, encephalitis, brain, T cells, microglia, interferon, PLX5622

## Abstract

Herpes simplex virus 1 (HSV-1) infects the majority of the human population and can induce encephalitis, which is the most common cause of sporadic, fatal encephalitis. An increase of microglia is detected in the brains of encephalitis patients. The issues regarding whether and how microglia protect the host and neurons from HSV-1 infection remain elusive. Using a murine infection model, we showed that HSV-1 infection on corneas increased the number of microglia to outnumber those of infiltrating leukocytes (macrophages, neutrophils, and T cells) and enhanced microglia activation in brains. HSV-1 antigens were detected in brain neurons, which were surrounded by microglia. Microglia depletion increased HSV-1 lethality of mice with elevated brain levels of viral loads, infected neurons, neuron loss, CD4 T cells, CD8 T cells, neutrophils, interferon (IFN)-β, and IFN-γ. In vitro studies demonstrated that microglia from infected mice reduced virus infectivity. Moreover, microglia induced IFN-β and the signaling pathway of signal transducer and activator of transcription (STAT) 1 to inhibit viral replication and damage of neurons. Our study reveals how microglia protect the host and neurons from HSV-1 infection.

## 1. Introduction

Viral infections can induce inflammation and irreversible impairment of neurons in brains of patients with encephalitis. Understanding the pathogenesis of viral encephalitis, which can cause death or severe and permanent sequelae in patients, is needed. The initial interaction between viruses and cells triggers a rapid induction of innate immune responses. Brain-resident microglia function as the sentinels of innate immunity responding promptly to virus infections. Increases of microglia, leukocytes infiltrating from the periphery, and severe neuron loss are detected in brains of patients with viral encephalitis [1]. Studies suggest that microglia might act as a double-edge sword during viral encephalitis. Microglia can exert antiviral activities by inducing themselves or other cells to express interferons (IFNs) and IFN-stimulated genes (ISGs) through signaling pathways, such as signal transducer and activator of transcription (STAT) 1. Additionally, microglia are recruited to eliminate infected neurons by phagocytosis [2] and secret cytokines and chemokines to orchestra brain immunity [3]. However, the excessive responses of microglia have been implicated to damage neurons and aggravate encephalitis induced by Zika virus [4,5] or human immunodeficiency virus [5,6] in patients. 

Mouse models are used to investigate the pathogenesis of viral encephalitis. Studies identifying the role of microglia in viral encephalitis in vivo are feasible in recent years because methods for efficient and specific depletion of microglia become available. Microglia depend on the signaling of colony-stimulating factor 1 receptor (CSF1R) for survival, growth, and proliferation. A blockade of CSF1R by small molecules, such as PLX5622, efficiently depletes microglia in mouse brains. Studies using PLX5622 to deplete microglia show increases of mortality and viral burden in mice infected with various viruses, demonstrating that microglia play a protective role in virus-induced encephalitis [2,7,8,9]. Mouse reports, which address the protective mechanisms of microglia, show that microglia regulate brain immune responses during virus infections. The specific immune responses affected by microglia depletion are diverse and dependent on the virus. Microglia depletion decreases monocytes in pseudorabies virus-infected mice [2]. Microglia depletion diminishes the numbers and activation of CD4 T cells, fails to affect the number of CD8 T cells, and increases *IFN-γ*, *interleukin (IL)-6*, and *IL-10* mRNA in Theiler’s murine encephalomyelitis virus (TMEV)-infected mice [7]. However, another report showed that microglia depletion decreases regulatory T cells but increases CD8 T cells in TMEV-infected mice [10]. Microglia depletion reduces the levels of major histocompatibility complex class II (MHC-II), CD4 T cells, CD4 T cells expressing IFN-γ, regulatory T cells, and CD8 cells, but increases macrophages and mRNA levels of *IFN-α*, *IFN-β*, and *IL-6,* in mouse hepatitis virus (MHV)-infected mice [8]. Microglia depletion lowers the levels of activated virus-specific (CD69^+^) and cytotoxic (CD160^+^) CD8 T cells, monocytes/macrophages, and *IFN-β*, *IFN-γ*, *tumor necrosis factor (TNF)-α*, *inducible NO synthase (iNOS)*, *CD86*, and *CD68* mRNA, but increases CD8 T cells in West Nile virus (WNV)-infected mice [9]. Additionally, microglia depletion increases *CCL2*, *CCL3*, *CCL7*, *CXCL9*, and *CXCL10* mRNA in WNV-infected mice [11]. Accordingly, microglia can regulate the expression of IFNs (IFN-α, IFN-β, and IFN-γ) with direct antiviral activity and cytokines (IL-6, IL-10, and TNF-α) to modulate viral infections. 

Herpes simplex virus (HSV)-induced encephalitis is the most common cause of sporadic, fatal encephalitis with an incidence of 1 in 200,000 individuals per year [12,13]. Acyclovir-related nucleoside analogs are used for patient treatment. The mortality rates of untreated and treated patients are over 70% and 30%, respectively, and only 2.5% of all patients return to normal neurological functions [13]. HSV-1 and HSV-2 are two different species belonging to the *Simplexvirus* genus of *Herpesviridae* family [14]. HSV-1 infects the majority (>80%) of adults worldwide [12,15] and accounts for more than 95% of encephalitis cases in patients beyond the neonatal period [16]. In human brains with HSV-1 encephalitis, magnetic resonance imaging shows destruction in the temporal lobe region, in which viral antigens are detected in neurons, but not in microglia [2], suggesting that neurons, but not microglia, support virus replication. Remarkably, infected cells are surrounded by microglia with an average of 1.5 microglia/HSV-1-positive cells [2]. Numerous mouse studies show that endogenous IFN-β, IFN-γ, CD4 T cells, and CD8 T cells reduce HSV-1 lethality [17,18,19,20,21,22]. One recent study using the approach of depleting *CSF1R* in C57BL/6J-derived *CSF1R-loxP-CX3CR1-cre/ERT2* mice found that transient and incomplete depletion of microglia in the early stage of HSV-1 infection decreased the brain levels of *IFN-β* mRNA and infiltrating monocytes/macrophages, neutrophils, and T cells [23]. In the present study, we showed that HSV-1 increased the number, activation, and antiviral activity of microglia, the most abundant leukocytes, in brains of infected mice. More importantly, we employed PLX5622 to efficiently deplete microglia before and during the entire period of infection and found that depletion of microglia by this way increased HSV-1 lethality of mice with elevated brain levels of viral loads, infected neurons, neuron loss, CD4 T cells, CD8 T cells, neutrophils, IFN-β, and IFN-γ. Our results showing that microglia depletion increases many types of immune responses in HSV-1-infected mice are different from the previous HSV-1 report [23] and are rarely seen in other virus infections [2,7,8,9,10]. 

## 2. Results

### 2.1. HSV-1 Infection of Mice Increases the Number, Activation, and Antiviral Activity of Brain Microglia

We assessed brain microglia in vivo using mice infected with HSV-1 on the scarified cornea, as the virus can infect human corneas to induce diseases [12,24]. All infected mice survived, and virus was detected in mouse brains on days 5 and 7 post-infection (p.i.) (Figure 1A). The leukocytes in brains of mock-infected and infected mice were analyzed by flow cytometry with the gating strategies shown in Appendix A. In brains of mock-infected mice, the numbers of (CD45^int^CD11b^+^) microglia, (CD45^hi^CD11b^+^Ly6G^-^) macrophages, (CD45^hi^CD11b^+^Ly6G^+^) neutrophils, CD4 T (CD45^hi^CD4^+^) cells, and CD8 T (CD45^hi^CD8^+^) cells were low (Figure 1B). In brains of infected mice, the numbers of these five different leukocytes were increased on day 5 or 7 p.i., especially microglia, which were the most abundant leukocytes detected. We performed immunofluorescence staining on brains. HSV-1 antigens were detected in one of three brains in the cortex (temporal lobe region) mostly in the cells positive for NeuN, a marker of neuron nuclei, on day 5 p.i., showing that neurons are the HSV-1 target (Figure 1C). Moreover, HSV-1-infected neurons were surrounded by microglia (Iba1-positive cells) with an amoeboid shape on day 5 p.i. Reports of ours and others found HSV-1 antigens and infectious virus in the temporal lobe of infected humans and mice [12,25,26]; thus, we monitored the temporal lobe in our further immunofluorescence analyses. 

Both Iba1 and the amoeboid shape are markers of activated microglia; thus, we further analyzed the activation of microglia using CD68, which is a lysosome/phagosome marker expressed in activated microglia with elevated phagocytic activity [2]. In brains of mock-infected mice, there were few (CD45^int^CD11b^+^CD68^+^) activated microglia (Figure 1D). In brains of infected mice, the levels of activated microglia were low in the early stage of infection (on day 3 p.i.) and increased on day 5 p.i. Because HSV-1 infection increased microglia activation in mouse brains on day 5 p.i. as demonstrated by makers of CD68 and Iba1 as well as the amoeboid shape, we examined the antiviral activity of brain microglia by adding virus to microglia harvested from brains of mice mock-infected or infected with virus for 5 days. After co-culture of microglia with virus for 12 h, the samples were frozen and subjected to plaque assay to determine viral titers in samples. The amount of infectious virus detected in the culture with brain microglia harvested from infected mice was lower than that with brain microglia harvested from mock-infected mice (Figure 1E). Our result showing that activated microglia reduce HSV-1 infectivity is consistent with previous findings of robust expression of antiviral mediators, such as TNF-α and CXCL10, in both mouse and human primary microglia stimulated with HSV-1 during nonproductive infection [27,28], as TNF-α and CXCL10 reduce HSV-1 replication in permissive cells [27]. 

### 2.2. Depletion of Microglia by PLX5622 Increases HSV-1 Lethality, Tissue Viral Loads, and Brain Neuron Loss of Infected Mice

We depleted microglia by feeding mice with chow containing the compound PLX5622. To efficiently deplete microglia, we fed mice with chow two weeks before infection as previously described [9], and our flow cytometry results showed that PLX5622 treatment reduced the number of brain (CD45^+^CD11b^+^Tmem119^+^) microglia by 95%, when compared to control chow (data not shown), as previously described [9,29]. Without infection, mice treated with PLX5622 for four weeks appeared normal, and the mouse survival was unaffected by PLX5622 one month or longer after treatment. This is consistent with the previous report showing that the effect of PLX5622 on neurons is minimal [30]. Mice fed with chow for 2 weeks were infected with HSV-1 and treated with chow during the entire observation period, day 14 p.i. The death rate of PLX5622-treated mice (88%, 15 of 17) was higher than that of control mice (0%, 0 of 16) on day 14 p.i. (Figure 2A). In control mice, virus was detected only in the brain, trigeminal ganglion, and eye, in which microglia are present [5,31,32] and likely function to reduce viral titers and antigen expression as well as virus-induced damage in tissues, especially in the mouse brain to protect mice from fatal infection. To address this issue, we first monitored tissue viral loads. The mean viral titers in brains and eyes of PLX5622-treated mice were higher than those of control mice on both days 5 and 7 p.i., and the mean viral titer in trigeminal ganglia of PLX5622-treated mice was higher than that of control mice on day 5 p.i. (Figure 2B). 

We performed immunofluorescence staining on brains of mice infected for 7 days, at the time when there was a huge (about 100-fold) difference in brain viral titers between control and PLX5622-treated groups, when compared to day 5 p.i. In the temporal lobe of control mice, neurons (NeuN-positive cells) were surrounded by microglia (Iba1-positive cells), and HSV-1 antigens were hardly detected (Figure 2C). HSV-1 antigens were detected in brain neurons of control mice on day 5 p.i. (Figure 1C), but not on day 7 p.i. (Figure 2C), probably because the brain viral titer was reduced by 1.7 log on day 7 p.i. when compared to day 5 p.i. (Figure 1A). In the temporal lobe of all three PLX5622-treated mice examined, abundant HSV-1 antigens were detected in neurons. Cleaved caspase 3 (an apoptosis marker) was detected in neurons of PLX5622-treated mice, but not in those of control mice (Figure 2D,E), showing that apoptosis, an indicator of damage, was detected in brain neurons of PLX5622-treated mice. We quantified the numbers of microglia (Iba1-positive cells) and neurons (NeuN-positive cells) in the temporal lobe (Figure 3A–C) and brain stem (Figure 3D,E). PLX5622 treatment reduced the number of microglia (Iba1-positive cells) by 80% (Figure 3B). Microglia of control mice displayed the ramified shape normally seen in the healthy brain [33,34], but microglia of PLX5622-treated mice showed the amoeboid or round shape normally found in the brain with neuron damage (Figure 3A,D). PLX5622 treatment reduced the numbers of neurons (Figure 3A,C), the number and length of neuron dendrites (Figure 3D), and the cell body size of neurons (Figure 3D,E), which were surrounded by the round shape of microglia (Figure 3D). 

### 2.3. PLX5622 Decreases Microglia, but Increases the Levels of Neutrophils, CD4 T Cells, CD8 T Cells, IFN-β, and IFN-γ in Brains of Infected Mice

Microglia serve as resident antigen-presenting cells to orchestrate immune cell infiltrating into brains [3,5,35]; thus, PLX5622 treatment might compromise the immune responses in brains of infected mice. We monitored the effect of PLX5622 treatment on mouse brain leukocytes using flow cytometry. In brains of infected mice, PLX5622 treatment reduced the number of (CD45^int^CD11b^+^) microglia by 80% and the antigen-presenting activities of microglia, including the levels of cells expressing MHC-II and the co-stimulating molecule (CD86), on day 5 p.i. (Appendix AA–C). We also monitored other myeloid cells, macrophages, and neutrophils, which infiltrate into the brain to regulate HSV-1 infection [36]. PLX5622 treatment slightly and significantly increased the numbers of infiltrating (CD45^hi^CD11b^+^Ly6G^−^) macrophages (Appendix A) and (CD45^hi^CD11b^+^Ly6G^+^) neutrophils (Figure 4A), respectively, on day 7 p.i. We assessed the effect of PLX5622 treatment on leukocyte activation using the marker, MHC-II. The levels of antigen-presenting (CD45^+^MHC-II^+^) cells and (CD45^hi^MHC-II^+^CD11b^+^Ly6G^−^) macrophages in infected mice with or without PLX5622 treatment were statistically insignificant on days 5 and 7 p.i. (Appendix A), suggesting that PLX5622 treatment fails to reduce the activation of non-microglia leukocytes in brains of infected mice. Overall, PLX5622 treatment specifically reduces the number and antigen-presenting activities of microglia, but fails to decrease the levels of infiltrating macrophages and neutrophils as well as antigen presenting cells and macrophages, in brains of infected mice. 

Microglia regulate T cell activities in brains of mice infected with TMEV, MHV, or WNV [7,8,9,10]. As T cells are important anti-HSV-1 effectors [18,19], we studied the effect of microglia depletion on the brain levels of infiltrating T (CD45^hi^CD3^+^) cells, which were low on day 5 p.i. and became detectable on day 7 p.i. in infected mice with or without PLX5622 treatment (Figure 4B). We therefore measured brain CD4^+^ T cells on day 7 p.i. and found that microglia depletion increased the number of CD4^+^ T cells (Figure 4C). This result is consistent with our finding of comparable MHC-II levels detected on leukocytes in brains of mice with or without microglia depletion (Appendix AE,F). Our further analysis showed that microglia depletion failed to significantly enhance the level of CD4^+^ T cells expressing IFN-γ (IFN-γ^+^CD4^+^ cells), which could be T helper type 1 (T_H_1) cells, on day 7 p.i. (data not shown). 

We also analyzed CD8 T cells in brains of infected mice, as the cells are one of the anti-HSV-1 effectors [17]. The levels of CD8^+^ T cells in infected mice with or without PLX5622 treatment were low on day 5 p.i. (Figure 5A). Microglia depletion increased the number of CD8^+^ T cells with about two-fold more CD8^+^ T cells than CD4^+^ T cells on day 7 p.i. (Figure 4C and Figure 5A). Microglia depletion increased T cells, especially CD8 T cells, but the viral load remained high, in mouse brains on day 7 p.i. We therefore determined whether microglia depletion compromises the activation of CD8 T cells. The levels of early activated CD8 (CD45^hi^CD3^+^CD8^+^CD69^+^) T cells and cytotoxic CD8 (CD45^hi^CD3^+^CD8^+^ CD160^+^) T cells in brains of mice with microglia depletion were slightly higher than those of control mice (Figure 5B,C). Activated CD8 T cells can suppress HSV-1 replication in mouse neurons [37] through IFN-γ and granzyme B [38]. We evaluated and found that microglia depletion enhanced the level of CD8 T cells expressing IFN-γ (IFN-γ^+^CD8^+^ cells) (Figure 5D–F) and slightly increased the levels of activated CD8 T cells expressing granzyme B (CD8^+^CD69^+^granzyme B^+^ cells), HSV-1-specific (CD8^+^gB tetramer^+^) CD8 T cells, and activated virus-specific CD8 (CD8^+^gB tetramer^+^CD69^+^) T cells in brains of infected mice (Figure 5G–I). We also assessed the capacity of brain CD8 T cells to suppress HSV-1 replication in brain neurons using an in vitro assay. Primary neurons cultured from brains of mouse embryos and infected with virus for 48 h produced virus. Co-culture of brain CD8^+^ T cells from infected mice with or without microglia depletion reduced viral titers of mouse primary brain neurons to the same degree by 50% (Figure 5J). Overall, microglia depletion increases the levels of CD8 T cells and CD8 T cells expressing IFN-γ, but not the anti-HSV-1 activity of CD8 T cells. In mock-infected mice treated with or without PLX5622, the brain levels of leukocytes, neutrophils, CD4 T cells, CD8 T cells, and IFN-γ^+^CD8^+^cells were below detection. Accordingly, the significant increases in brain levels of these four types of leukocytes in infected mice treated with PLX5622 should be the effect of HSV-1 infection in combination with PLX5622. 

We measured IFN-γ in brains of infected mice on day 7 p.i. by ELISA, as microglia depletion enhanced the population of IFN-γ^+^CD8^+^ T cells at this time point. Microglia depletion increased IFN-γ in brains of infected mice (Figure 6A). We also measured IFN-β and found that microglia depletion increased IFN-β in brains of infected mice on both days 5 and 7 p.i. (Figure 6B). In brains of mock-infected mice treated with or without PLX5622, both IFN-β and IFN-γ levels were below detection. We determined whether the increases of IFNs can induce the expression of ISGs, *CXCL10* and *MX dynamin-like GTPase 1 (Mx1)*, by quantitative RT-PCR. Microglia depletion significantly and slightly increased *CXCL10* and *Mx1* mRNA levels, respectively in brains of infected mice on day 5 p.i. (Figure 6C). We also monitored the mRNA levels of pro-inflammatory modulators, IL-1β, IL-6, iNOS, and TNF-α, as well as anti-inflammatory modulators, arginase-1, IL-10, and transforming growth factor (TGF)-β, in brains of infected mice on day 5 p.i. (Figure 6D). Among these, *IL-6*, *iNOS*, *TNF-α*, and *IL-10* mRNA levels are reported to be affected by microglia depletion in mice infected with other viruses [7,9]. Microglia depletion slightly increased *IL-1β* and *IL-6* mRNA levels and failed to significantly affect the expression of other immune modulators. In brains of mock-infected mice treated with or without PLX5622, the levels of 11 immune modulators listed in Figure 6 were below detection. Collectively, microglia depletion provokes an inflammatory environment in brains of infected mice.

### 2.4. Mouse Primary Brain Microglia Inhibit HSV-1 Replication of Mouse Primary Brain Neurons with Increases of IFN-β Expression and STAT1 Phosphorylation

Our in vivo results showed that microglia reduce the number of HSV-1-infected neurons in mouse brains (Figure 2C). We further searched how microglia protect neurons from HSV-1 using in vitro study with mouse primary cells. The issue regarding whether microglia decrease HSV-1 replication in neurons in vitro is controversy. One report showed the failure of primary microglia harvested from uninfected adult mice to suppress HSV-1 replication in a mouse neuronal cell line [39]. However, the other report showed that HSV-1 induces IFN-β in mouse primary brain microglia, but not in mouse primary neurons [3]. The same report further showed that pretreatment of neurons with the condition medium from infected microglia, but not from uninfected microglia, before HSV-1 infection reduces the viral production with enhanced expression of an ISG *(CXCL10)* mRNA. In the brain, microglia and neurons are in contact, we therefore performed the co-culture study of these two cells to address the controversy of whether microglia can inhibit HSV-1 replication and virus-induced damage of neurons. We cultured primary microglia from brains of neonatal mice and primary neurons from brains of mouse embryos and monitored HSV-1 replication in the cells. The viral titers of microglia infected with a multiplicity of infection (MOI) of 1 decreased from 36 to 48 h p.i. and were below detection 72 h p.i. (Figure 7A). This finding is consistent with previous reports showing that both primary mouse and human microglia fail to support HSV-1 replication [27,28]. However, the viral titers of neurons infected with a low MOI (0.001) increased from 36 to 72 h p.i. (Figure 7B). Accordingly, microglia suppress HSV-1 replication, but neurons support HSV-1 replication. Neurons were then incubated without or with microglia at a ratio of 1:1, infected with HSV-1 (MOI = 0.001), and harvested 72 h p.i. for virus titration. Infected neurons produced a high viral titer, and co-culture of microglia with neurons reduced the viral titer by 47% (Figure 7C). Western blotting results showed high levels of HSV-1 essential protein (ICP8) and cleaved caspase 3 in infected neurons, and co-culture of microglia with neurons reduced ICP8 and cleaved caspase 3 levels (Figure 7D–F). 

As microglia suppress HSV-1 replication and apoptosis of infected neurons, we studied the signaling of antiviral action. Microglia can secret the antiviral cytokines, type I IFN (IFN-β) and type II IFN (IFN-γ), which can inhibit HSV-1 replication by activating (phosphorylating) the transcription factor, STAT1 of ISGs [40,41]. We monitored type I IFN production and found that infected neurons failed to produce IFN-β 72 h p.i. (Figure 8A) in a manner consistent with a previous study using the mouse primary neurons cultured from an unidentified neural tissue [3]. However, co-culture of microglia with neurons induced IFN-β production 72 h p.i. STAT1 is shown to protect mice from HSV-1 infection, as *Stat1*^-/−^ mice are highly susceptible to infection with elevated tissue viral loads and mortality [42]. Furthermore, the conditional knockout of *Stat1* in neural stem cells, progenitors, and glia increased HSV-1 lethality and tissue viral loads of mice [43]. We studied the significance of STAT1 for microglia to inhibit HSV-1 replication of neurons in co-culture. We first monitored STAT1 phosphorylation (p-STAT1), which was in a level extremely low in infected neurons and was induced in the co-culture of microglia with neurons, 72 h p.i. (Figure 8B–D). To assess the significance of STAT1 in reducing the viral yield, we cultured primary neurons and microglia from brains of wild-type or *Stat1*^−/−^ mice. Co-culture of wild-type microglia with wild-type neurons, but not with *Stat1*^−/−^ neurons, reduced the viral titer (Figure 8E), showing that the STAT1 signaling of neurons is required to suppress HSV-1 replication. Co-culture of *Stat1*^−/−^ microglia with wild-type or *Stat1*^−/−^ neurons failed to reduce viral titers, showing that microglia exert anti-HSV-1 activity via STAT1. We further searched and found that the capacity of *Stat1*^−/−^ microglia to produce IFN-β was severely impaired in co-culture with neurons (Figure 8F) in a way consistent with previous reports showing that IFN-β activates STAT1 to increase its own production in an autocrine manner in a mouse microglia cell line [40,44]. Results of these in vitro studies resolve the controversy of whether microglia protect neurons from HSV-1 infection and provide the protective mechanism. Our additional results showed that co-culture of microglia with neurons failed to induce IFN-γ and STAT3 activation (data not shown).

## 3. Discussion

The present study reveals how microglia protect the host and neurons from HSV-1 infection. In mice, HSV-1 infection increases the number and activation of brain microglia. Depletion of microglia before and during infection augments virus lethality of mice with elevated brain levels of viral loads, infected neurons, neuron loss, CD4 T cells, CD8 T cells, neutrophils, IFN-β, and IFN-γ, showing that microglia exert protective effect. We performed an in vitro study to provide mechanistic insights on how microglia are protective and found that brain microglia from infected mice reduce virus infectivity and that brain microglia induce IFN-β and STAT1 activation to inhibit viral replication and damage of brain neurons. Numerous reports show that endogenous CD4 T cells, CD8 T cells, IFN-β, and IFN-γ protect mice from HSV-1 infection [17,18,19,20,21,22]. Microglia depletion increases brain levels of CD4 T cells, CD8 T cells, IFN-β, and IFN-γ. However, these compensatory leukocyte and cytokine responses are futile to reduce brain viral loads and virus lethality. Here, we add microglia into the list of effectors protecting mice from infection and highlight the significance of microglia.

The recent HSV-1 study of Uyar et al. showed that transient and incomplete depletion of microglia during early stage of infection decreased the brain levels of *IFN-β* mRNA and infiltrating monocytes/macrophages, neutrophils, and T cells [23] in manners contrast to our study with long term and efficient microglia depletion via PLX 5622 before and during the entire infection period. Studies of Uyar et al. and ours showed that microglia depletion increased brain viral titers by about 10- and 100-fold on days 6–7 p.i. and mortality rates of infected mice by 60 and 85%, respectively. Studies of Uyar et al. and ours showed the effects of different degrees and timing of microglia depletion on the responses of IFN-β, monocytes/macrophages, neutrophils, and T cells in the brain of infected mice. In brains, microglia depletion via PLX 5622 decreases the number of CD4 T cells in TMEV-infected mice [7], the levels of CD4 T cells, CD4 T cells expressing IFN-γ, and CD8 T cells in MHV-infected mice [8], the levels of activated virus-specific and cytotoxic CD8 T cells as well as *IFN-β*, *IFN-γ*, *TNF-α*, *iNOS*, *CD86*, and *CD68* mRNA in WNV-infected mice [9]. Accordingly, microglia may collaborate with CD4 T cells to protect TMEV-infected mice, with CD4 T cells, CD8 T cells, and IFN-γ to protect MHV-infected mice, with activated virus-specific and cytotoxic CD8 T cells as well as IFN-β, IFN-γ, TNF-α, and iNOS to protect WNV-infected mice. Here, we show that brain microglia decrease HSV-1 infectivity and reduce virus replication of brain neurons. Microglia depletion increases HSV-1 lethality of mice with elevated levels of CD4 T cells, CD8 T cells, neutrophils, IFN-β, and IFN-γ. These results suggest that microglia exert anti-HSV-1 activity efficiently. The issues regarding whether the abundance and antiviral activity of brain microglia are high in HSV-1-infected mice, when compared to mice infected with TMEV, MHV, or WNV need further investigation. In our study, the survival rates of HSV-1-infected mice with microglia depletion were 100%, 60%, and 12% on days 5, 7, and 12 p.i., respectively. In these mice, elevated (IFN-β) immune responses were detected as early as day 5 p.i., and high levels of IFN-β, neutrophils, CD4 T cells, CD8 T cells, and IFN-γ were detected on day 7 p.i., showing that the enhanced immune responses mediated by microglia depletion occur before and during mice succumb to death and should not be the abnormal responses that only occur in a subset of mice and correlate with prolonged survival. In brains of HSV-1-infected mice with microglia depletion, the elevated levels of CD4 T cells, CD8 T cells, neutrophils, IFN-β, and IFN-γ should be triggered by elevated viral loads on day 7 p.i. 

A recent study showed that persistent HSV-1 infection of neurons triggers (CD3^+^IFN-γ^+^CD8^+^) CD8 T cell response and gastrointestinal neuromuscular dysfunction in the enteric nervous system of mice after orogastric inoculation [45]. This study further found that infiltrating CD8 T cells leads to intestinal dysmotility during HSV-1 infection, as depletion of intestinal CD3^+^CD8^+^ cells by the anti-CD8 monoclonal antibody ameliorates intestinal dysfunction. Moreover, another report showed that T cells survey the ganglionic region containing latently infected neurons and participate in preventing reactivation of HSV-1 from latency whereas cytokines released by HSV-1-activated T cells have been suggested to contribute to neuronal damage [46]. Here, we showed that microglia depletion increases T cell and IFN responses in brains of infected mice. However, these compensatory responses fail to reduce brain viral loads and virus lethality of mice. The issue regarding whether the elevated T cell and IFN responses induced by microglia depletion are associated with pathology or damage detected in brains of HSV-1-infected mice needs further investigation. 

CSF1R is expressed by microglia and macrophages [47,48,49], but not by normal lymphocytes [50]. Thirty years of studies show that microglia, but not macrophages**,** are unusually dependent on CSF1R signals and that suppression of CSF1R in mice, rats, and humans impairs microglia, but spares many peripheral macrophage populations [51]. Consistently, most of reports, especially those addressing the roles of microglia in viral infections using the CSF1R inhibitor, PLX5622 showed that in uninfected mice, this pharmacological agent specifically depletes microglia, but fails to reduce (CD45^+^CD11b^+^) mononuclear phagocytes (macrophages and monocytes), macrophages (F4/80^+^ or Iba^+^ cells), neutrophils, CD4 T cells, and CD8 T cells in the periphery (spleen, blood, bone marrow, and/or lymph nodes) and monocytes, macrophages, lymphocytes, and CD8 T cells in the central nervous system, brain and/or spinal cord [2,7,8,9]. Only one recent report found that PLX5622 fails to decrease CD45^+^CD11b^+^ cells but reduces macrophages (F4/80^+^ cells), CD4 T cells, and CD8 T cells, in the bone marrow and CD8 T cells in the spleen of mice [52]. However, the reliability of this brief report was challenged, as the presented results are uninterpretable because of critical data missing [51]. In our murine infection model, HSV-1 is detected in the brain, trigeminal ganglion, and eye, but not in other tissues, organs, and blood. Our unpublished results showed minimal levels of (CD45^hi^CD11b^+^Ly6G^-^) macrophages, CD4 T cells, and CD8 T cells in the eyes, trigeminal ganglia, and brains of uninfected mice treated with or without PLX5622. Our results revealed that HSV-1 infection increased the numbers of these three types of leukocytes in mouse brains. Moreover, PLX5622 further increased the numbers of these three types of leukocytes in brains of HSV-1-infected mice in a manner unlikely due to PLX5622-mediated reduction of these three leukocytes in the periphery. To further investigate whether PLX5622 reduces the immune responses of the peripheral organ in our mouse model, we monitored the individual eye for mRNA levels due to the small size of eye. The mRNA levels of cytokines (IFN-β, IFN-γ, TNF-α, and IL-6), chemokine (CXCL10), infiltrating leukocyte marker (*CD45*), and T cell markers (*CD4* and *CD8*) in the eyes of uninfected mice treated with or without PLX5622 were below detection. In the eyes collected from infected mice on day 5 p.i., the levels of these mRNA were detectable, and microglia depletion slightly increased the levels of *IL-6,* and *CXCL10* mRNA (Appendix A). Our eye results suggest that the elevated immune responses in brains of infected mice with PLX5622 treatment are due to increases in local viral titer/inflammation and unlikely due to altered induction of the immune response. In the mouse brain, PLX5622 increases CD4 and CD8 T cells in HSV-1-infected mice, macrophages in MHV-infected mice [8], and CD8 T cells in mice infected with TMEV or WNV [9,10]. These studies provide a better understanding that PLX5622 fails to reduce the infiltration of macrophages and T cells into brains of infected mice. 

Our in vivo results show that microglia reduce brain viral titers, infected neurons, and neuron apoptosis and that HSV-1 infection activates microglia to express CD68, a marker for lysosome/phagosome and phagocytic activity. We therefore performed an in vitro study to provide mechanistic insights on how microglia reduce HSV-1. We found that activated adult microglia from infected mice decrease HSV-1 infectivity. We also searched how microglia protect neurons. HSV-1 is known to induce IFN-β in microglia, but not in neurons, via the cGAS-STING sensing pathway in mice [3]. However, the issue regarding whether microglia decrease HSV-1 replication in neurons in vitro is controversy. Here, we show that brain microglia induce the IFN-β-STAT1 signaling pathway to suppress viral replication and apoptosis of brain neurons in a mechanism not reported before. Overall, our in vitro study provides two protective mechanisms of microglia. Microglia reduce HSV-1 infectivity and protect neurons from HSV-1 infection. In the result showing that microglia reduce HSV-1 infectivity, we isolated activate microglia from infected (adult) mice for study. Because culture of microglia from neonate mice provides a 50- to 10-fold higher yield of cells (1 × 10^6^ cells/mouse brain) than isolation of microglia from adult mice (0.2 to 1 × 10^5^ cells/mouse brain), neonatal microglia are conventionally used for studies [53,54]. We used neonatal microglia for IFN-β and STAT1 study to save tens and hundreds of adult mice. Our in vitro results showed that both adult and neonatal microglia exert anti-HSV-1 effect. Infected control mice with low levels of IFN-β on days 5 and 7 p.i. survived, but infected mice with microglia depletion and elevated IFN-β levels succumbed to death. Future in vivo studies are needed to clarify issues regarding what IFN-β-producing cells protect neurons from infection, whether IFN-β-producing microglia protect neurons from infection, and whether IFN-β-producing microglia are more effective than other IFN-β-producing cells in protecting neurons from infection. 

We used C57BL/6J mice for in vivo study because we previously showed that this mouse strain supports the replication of all tested HSV-1 strains (294.1, McKrae, KOS, and RE) with high brain viral titers on day 5 p.i. [55]. Strains 294.1 and McKrae, but not KOS and RE, continued to produce high levels of viral titers in the brain on day 7 p.i. and induced death in infected mice. This result is consistent with that of present study showing that microglia depletion increases brain viral titers on both days 5 and 7 p.i. to induce mouse death. Another major reason to use C57BL/6J mice for in vivo studies is to be consistent with in vitro studies, which use primary brain cells cultured from C57BL/6J-derived *Stat1*^−/−^ mice. We used 6- to 8-week-old adult mice for study to address the phenomenon observed in humans that HSV-1 accounts for most adult encephalitis cases worldwide. In both murine infection model and humans, age, sex, and the dose of virus inoculum may affect HSV-1 lethality. Future studies to vary age, sex, and the dose of virus inoculum in the murine infection model could provide information for the impact of these factors on how microglia exert anti-HSV-1 activity to protect humans from fatal encephalitis. 

## 4. Materials and Methods

### 4.1. Cells, Virus, and Mice

The Vero cell line was maintained and propagated according to the instructions of the American Type Culture Collection. HSV-1 strain 294.1 was propagated and titrated on Vero cell monolayers and used for studies. C57BL/6J mice were purchased from the National Laboratory Animal Center of Taiwan. C57BL/6J-derived *Stat1* knockout (B6.129S (Cg)-*Stat1*^tm1Dlv^/J) mice with the targeted disruption of gene were kindly provided by Dr. Chien-Kuo Lee in National Taiwan University. The mice were bred and maintained under specific-pathogen-free conditions in the laboratory animal center of our college. 

### 4.2. Infection and Microglia Depletion of Mice

Six- to eight-week-old female C57BL/6J mice were anesthetized and inoculated with 294.1 (2 × 10^5^ plaque forming units) on one eye after scarification of the cornea with a needle 10–12 times. Mouse eyeballs, ipsilateral trigeminal ganglia of infected eyes, and whole brains were collected, placed in the tube with 1 mL medium, frozen, homogenized with tissue grinders, frozen, sonicated, and subjected to plaques assay to determine viral titers as previously described [55]. For the plaque assay, virus in solution was inoculated onto the Vero cell monolayer seeded the day before. After incubation for 45 min (min) at room temperature, the infected cell monolayer was overlaid with medium containing 1% methylcellulose, cultured for three more days, and stained with crystal violet to detect plaques. To deplete microglia, four-week-old female mice were fed with the chow containing the compound, PLX5622 (Plexxikon Inc., CA, USA) at the dose of 1200 mg/kg chow or control chow (Research Diet Inc., NJ, USA) two weeks before infection and during infection

### 4.3. Flow Cytometry 

The assay was performed as previously described [56]. Briefly, mouse brains were dissociated in homogenizers, and leukocytes were isolated by 30% and 70% stock isotonic Percoll (GE Healthcare, IL, USA) solution gradient centrifugation. Freshly isolated leukocytes were blocked with the antibody CD16/CD32 (clone 93; BioLegend, CA, USA) for 20 min to prevent nonspecific binding before staining with fluorescence dye-conjugated isotype antibodies or antibodies against mouse leukocyte antigens, CD45 (clone 30F11; BD Biosciences, NJ, USA), CD11b (clone M1/70; BD Biosciences, NJ, USA), Ly6G (clone 1A8; BD Biosciences, NJ, USA), CD3e (clone 145-2C11; BD Biosciences, NJ, USA), CD4 (clone GK-1.5; BioLegend, CA, USA), CD8 (clone 53-6.7; BioLegend, CA, USA), CXCR3 (clone CXCR3-173; eBioscience, Thermo Fisher Scientific, MA, USA), CD69 (clone H1.2F3; BioLegend, CA, USA), or CD160 (clone 7H1; BioLegend, CA, USA) for 45 min on ice in the dark. For detection of cell viability, leukocytes were stained with the fixable viability dye (Invitrogen, Thermo Fisher Scientific, MA, USA) for 30 min on ice in the dark. For detection of CD68, leukocytes were fixed and permeabilized by Cytofix/Cytoper (BD Biosciences, NJ, USA) for 20 min on ice before incubation with the antibody against mouse CD68 (clone FA-11; BioLegend, CA, USA) for 45 min on ice in the dark. IFN-γ is transiently expressed and difficult to be detected in CD8 T cells in vivo. In order to recapitulate and quantify CD8 T cells expressing IFN-γ, phorbol 12-myristate 13-acetate (PMA) (MilliporeSigma, Merck, MA, USA) stimulation has been the gold standard used [57]. For detection of IFN-γ^+^, granzyme B^+^, and HSV-1-specific T cells, leukocytes were stimulated by PMA (0.081 μM) and ionomycin (1.34 μM), treated with GolgiPlug (BD Biosciences, NJ, USA) for 5 h at 37°C, fixed, and permeabilized by Cytofix/Cytoper for 20 min on ice before incubation with anti-IFN-γ antibody (clone XMG1.2; BioLegend, CA, USA), anti-granzyme B antibody (clone QA16A02; BioLegend, CA, USA), and the tetramer H-2K(b) HSV-1 gB_498-505_ (SSIEFARL) [58] labeled with brilliant violet 421 (generated and kindly provided by the NIH Tetramer Core Facility, GA, USA) for 45 min on ice in the dark. The stained cells were subjected to the flow cytometer CantoII (BD Biosciences, NJ, USA) and analyzed by FlowJo (Tree Star Inc, OR, USA).

### 4.4. Immunofluorescence Staining

Mouse brains were harvested, processed, and stained with DAPI dye (Abcam, Cambridge, UK) for DNA and antibodies against mouse NeuN (clone A60; Merck Millipore, MA, USA), cleaved caspase 3 (Cell Signaling Technology, MA, USA), or Iba1 (GeneTex, CA, USA), or HSV-1 (Bio-Rad, CA, USA) as previously described [59]. Images were photographed by the confocal laser scanning microscope, FV3000 (Olympus, Tokyo, Japan), and Iba- or NeuN-positive cells were quantified by ImageJ (National Institutes of Health, NYC, USA).

### 4.5. Isolation of Microglia or CD8 T Cells from Mouse Brains

Freshly isolated leukocytes obtained by centrifugation as described in the flow cytometry were stained with the antibody against mouse P2RY12-conjugated with magnetic particles for the positive selection of microglia using the MojoSort^TM^ mouse P2RY12 selection kit (BioLegend, CA, USA) as previously described [60] or stained with antibodies against mouse CD4, CD11b, CD11c, CD19, CD24, CD45R/B220, CD49b, CD105, MHC-II, TER-119/Erythroid, or TCR-γδ conjugated with magnetic particles for the negative selection of CD8 T cells using the MojoSort^TM^ mouse CD8 T cell isolation kit (BioLegend, CA, USA) as previously described [61]. The purity of isolated microglia and CD8 T cells was >95% and >90%, respectively based on flow cytometric results of CX3CR1 and P2RY12 for microglia and CD8 for T cells, respectively.

### 4.6. Quantification of Cytokines

Mouse brains and cell cultures were harvested and placed in tubes with 1 mL of medium. Brains were frozen at −80 °C and homogenized. Homogenized brains and cell culture samples were centrifuged to obtain supernatants, which were subjected to enzyme-linked immunosorbent assays (ELISA) with commercially available kits according to the manufacturer’s instructions for detection of IFN-γ and IFN-β (BioLegend, CA, USA) with the detection limits of 8 and 2 pg/mL, respectively. 

### 4.7. RNA Isolation and Quantitative RT-PCR

Mouse brains were cut into half and frozen in liquid nitrogen, and total RNA was extracted by the RNeasy lipid tissue mini kit (Qiagen, Hilden, Germany). After reverse transcription with the reverse primers shown in Appendix A, the synthesized cDNA was used for quantitative PCR with the forward and reverse primers. PCR was performed with initial activation for 10 min at 95 °C followed by 40 cycles of denaturation (95 °C, 15 s) and annealing (60 °C, 1 min) with Fast SYBR Green Master Mix (Thermo Fisher Scientific, MA, USA). The threshold cycle (CT) of each product was determined, normalized to the internal control β-actin, and shown as ΔCT. All results are shown as the ratio to β-actin calculated as 2-ΔCT, and the means of control groups (infected mice without PLX5622 treatment) were set as 1.

### 4.8. Culture of Mouse Primary Cells

Primary neurons were isolated from the cortex and hippocampus of mouse embryos at 18.5 days post-coitum as previously described [62]. Briefly, mouse neural tissues were digested with papain (MilliporeSigma, Merck, MA, USA) for 15–20 min at 37 °C and dissociated into single cells in Neurobasal medium (Invitrogen, Thermo Fisher Scientific, MA, USA) containing 2 mM L-glutamine and 2% B-27 (Invitrogen, Thermo Fisher Scientific, MA, USA) by pipetting. After centrifugation, cells were seeded onto 24-well plates coated with poly-D-lysine (Invitrogen, Thermo Fisher Scientific, MA, USA) with Neurobasal medium at a density of 1 × 10^5^ cells per well. Primary neurons cultured for 5 days were used for experiments. Primary microglia were isolated from the mouse cortex at postnatal days 1 or 2 as previously described [53,54]. Briefly, mouse cortices were dissociated and suspended in DMEM/F12 medium (Invitrogen, Thermo Fisher Scientific, MA, USA) containing 10% fetal bovine serum (Invitrogen, Thermo Fisher Scientific, MA, USA). Cells were then cultured in flasks coated with poly-D-lysine. After 12–14 days, microglia were detached from mixed glial culture by the shake-off method and seeded onto 24-well plates coated with poly-D-lysine. The purity of neurons and microglia was >95% based on the observation of cell morphology.

### 4.9. Western Blotting

Cells were harvested, washed, frozen, and lysed. Total proteins were extracted from samples and measured for concentrations. Proteins were separated by polyacrylamide gel electrophoresis, blotted onto membranes (Merck Millipore, MA, USA), and blocked with 5% skim milk to prevent nonspecific binding. Blots were stained with antibodies against ICP8 (clone 11E2; Santa Cruz Biotechnology, TX, USA), cleaved caspase 3 (Cell Signaling Technology, MA, USA), phospho-STAT1 (Tyr701; clone 58D6; Cell Signaling Technology, MA, USA), STAT1 (Cell Signaling Technology, MA, USA), or β-actin (clone AC-15; MilliporeSigma, Merck, MA, USA) at 4 °C overnight and incubated with the corresponding HRP-conjugated secondary antibodies for chemiluminescent detection. Images were photographed by UVP BioSpectrum AC imaging system (Ultra-Violet products Ltd., Cambridge, UK) and analyzed by ImageJ (National Institutes of Health, NYC, USA). 

### 4.10. Statistical Analyses 

Data are expressed as means ± SEM (error bar) values. Data of Kaplan–Meier survival curves (Figure 2A) were analyzed by the log-rank test. Data of flow cytometry, viral titration, and ELISA analyses (Figure 1B,D,E, Figure 2B, Figure 4A,C, Figure 5A–C,E,F,J, Figure 6A,B, Figure 7C, and Figure 8A,E,F) were analyzed by the Mann–Whitney *U* test. The rest of the data were analyzed by the Student’s *t* test. GraphPad Prism 6.0 (GraphPad Software, 6.0, CA, USA) was used for statistical analyses. A *p* value of <0.05 was considered statistically significant. 

## Figures and Tables

**Figure 1 ijms-22-12457-f001:**
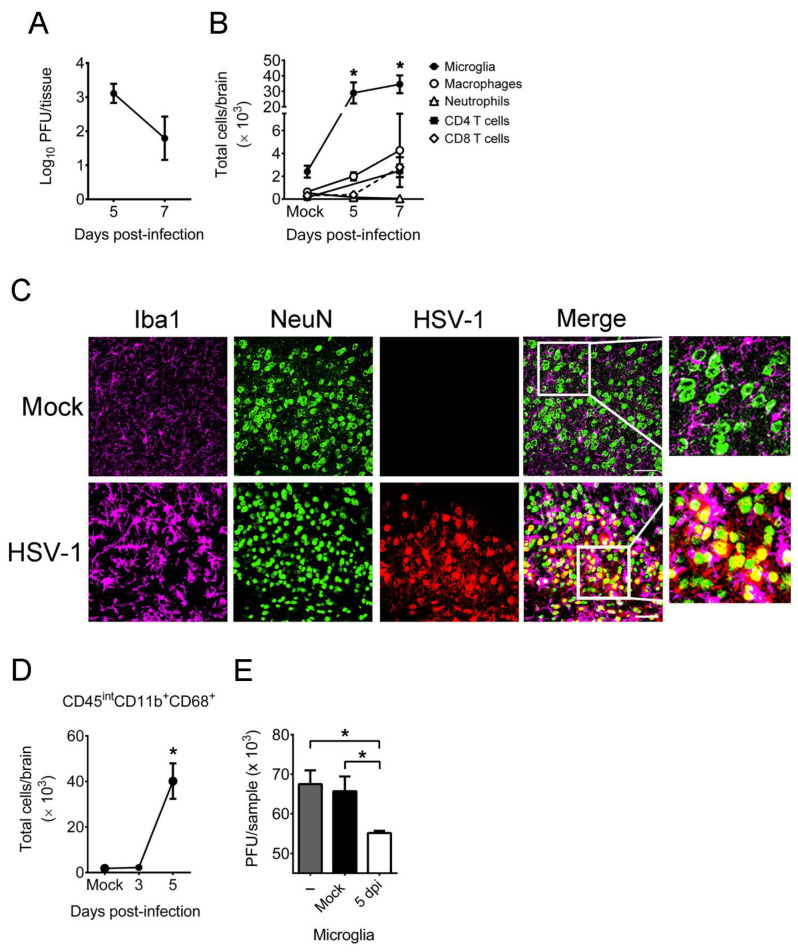
HSV-1 infection of mice increases the number and activation of brain microglia, which reduce HSV-1 infectivity. (**A**) Mouse brains were harvested on the indicated days for virus titration. (**B**) Brains of mock-infected and infected mice were assayed for markers of microglia (CD45^int^CD11b^+^), macrophages (CD45^hi^CD11b^+^Ly6G^-^), neutrophils (CD45^hi^CD11b^+^Ly6G^+^), CD4 T cells (CD45^hi^CD4^+^), and CD8 T cells (CD45^hi^CD8^+^) by flow cytometry. (**C**) Brains from mock-infected and infected mice harvested on day 5 post-infection were sectioned and subjected to immunofluorescence staining with antibodies against Iba1, NeuN, or HSV-1. The temporal lobe region is shown. Scale bar, 50 μm. Data are representative of at least 3 samples per group from two independent experiments. (**D**) The brains of mock-infected and infected mice were assayed for markers of activated microglia (CD45^int^CD11b^+^CD68^+^) by flow cytometry. (**E**) HSV-1 (MOI = 1) was incubated without (–) or with 1 × 10^5^ microglia isolated from the brains of mock-infected mice (Mock) or mice infected for 5 days (5 dpi) for 12 h and harvested for virus titration. The data represent means ± or + SEM (error bars) of 4–10 samples per data point obtained from at least two independent experiments in panels (**A**,**B**,**D**,**E**). *, *p* < 0.05, compared with the mock-infected group (**B**,**D**) or between the indicated groups (**E**).

**Figure 2 ijms-22-12457-f002:**
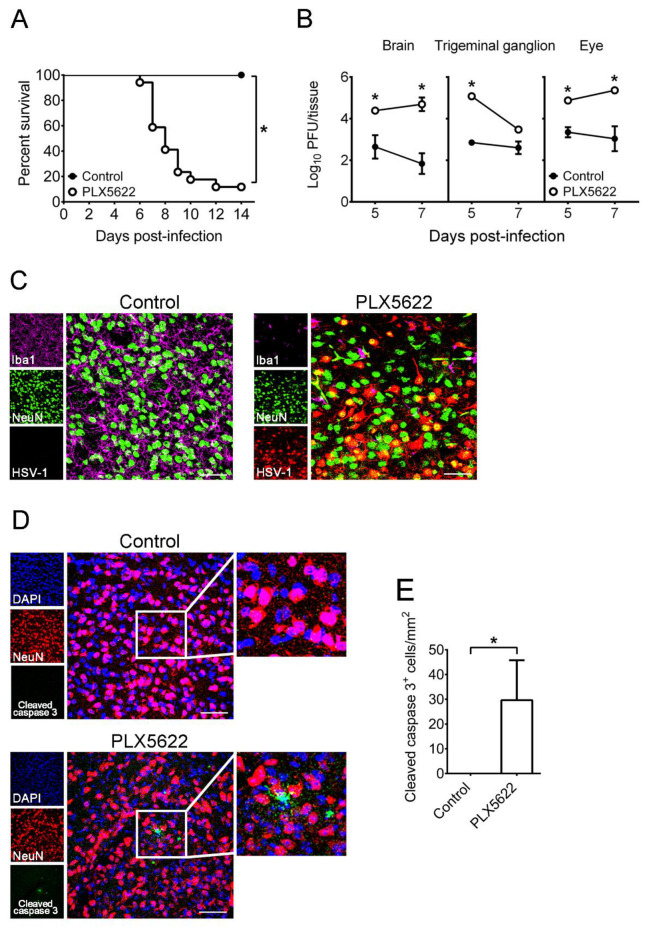
The effects of PLX5622 on HSV-1 lethality, tissue viral loads, and brain infected neurons as well as neuron apoptosis of mice. (**A**) Mice were fed with control chow (n = 16) or the chow containing PLX5622 (n = 17), infected with HSV-1, and monitored for survival. (**B**) The indicated organs and tissues of control and PLX5622-treated mice were harvested on the indicated days for virus titration and on day 7 post-infection for brain sections (**C**,**D**) before immunofluorescence staining with antibodies against Iba1, NeuN, HSV-1, or cleaved caspase 3. Scale bar, 50 μm. In panel **C**, right panels are the merged and magnified images of three left panels. In panel (**D**), middle panels are the merged and magnified images of three left panels, and right panels are the magnified images of indicated areas show in middle panels. (**E**) The number of leaved caspase 3^+^ cells in images (0.3 × 0.3 mm) were counted. Results shown in panels (**C**–**E**) are the temporal region and the representative of at least 3 samples per group from two independent experiments. The data represent means ± SEM (error bars) of 3–6 samples per data point obtained from at least two independent experiments in panels (**B**,**E**). *, *p* < 0.05, compared between the indicated groups (**A**,**E**) or with the control group on the same day (**B**).

**Figure 3 ijms-22-12457-f003:**
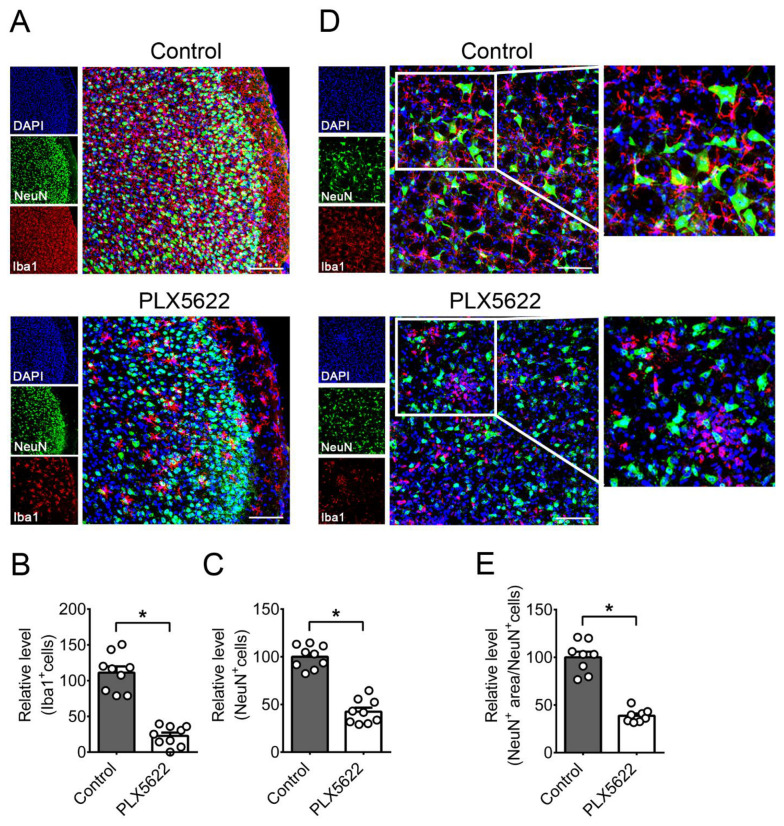
The effects of PLX5622 on the microglia and neurons in brains of infected mice. Sections of brains from infected mice fed with control chow or the chow containing PLX5622 were harvested on day 7 post-infection and subjected to immunofluorescence staining with antibodies against mouse NeuN or Iba1. The results of (**A**–**C**) temporal lobe and (**D**,**E**) brain stem regions are shown. Scale bar, 50 μm. In panel (**A**), right panels are the merged and magnified images of three left panels. In panel (**D**), middle panels are the merged and magnified images of three left panels, and right panels are the magnified images of indicated areas shown in middle panels. (**B**) Iba1^+^, (**C**) NeuN^+^ cells, and (**E**) NeuN^+^ area/NeuN^+^cells in the mouse temporal lobe in images (0.15 × 0.15 mm) were quantified. The means of control groups were set as 100%. The data represent means + SEM (error bars) of 3 mice per group obtained from at least two independent experiments with three or two sections per mouse. *, *p* < 0.05.

**Figure 4 ijms-22-12457-f004:**
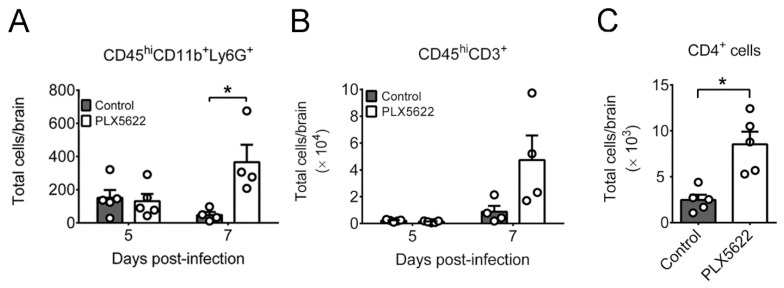
Effects of PLX5622 on leukocyte levels in brains of infected mice. Brains of infected mice fed with control chow or the chow containing PLX5622 were harvested on the indicated days or day 7 post-infection and assayed for (**A**) markers of neutrophils (CD45^hi^CD11b^+^Ly6G^+^), (**B**) T (CD45^hi^CD3^+^) cells, and (**C**) CD4^+^ T cells by flow cytometry. The data represent means + SEM (error bars) of 4-5 samples per group obtained from at least two independent experiments. *, *p* < 0.05.

**Figure 5 ijms-22-12457-f005:**
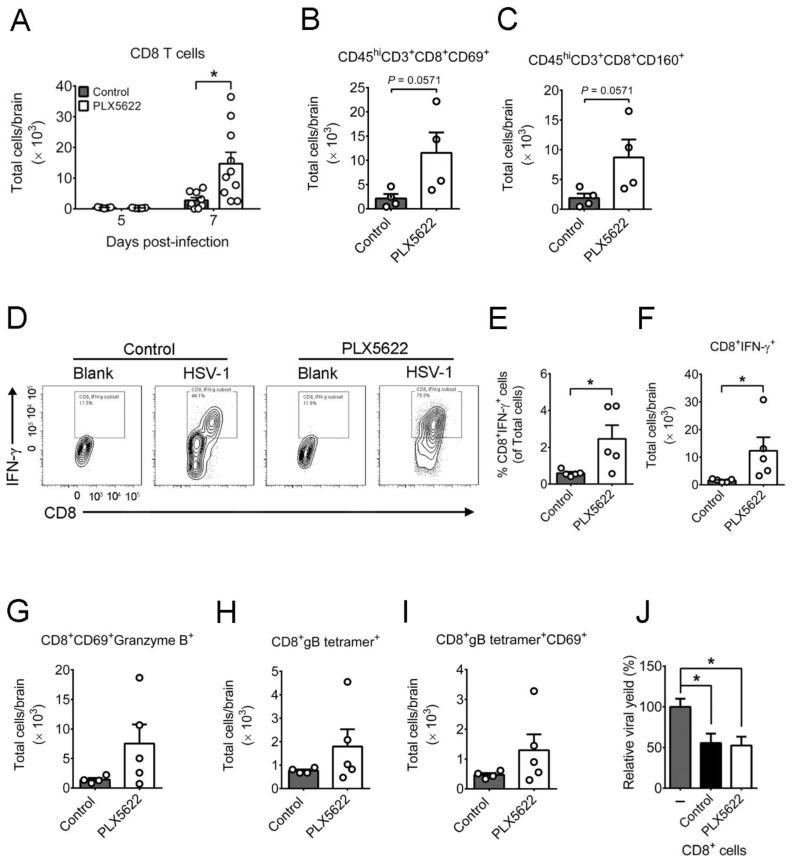
Effects of PLX5622 on the leukocyte responses in brains of infected mice. The brains of infected mice fed with control chow or the chow containing PLX5622 were harvested on the indicated days or day 7 post-infection and assayed for (**A**) CD8^+^ T cells, (**B**) early activated CD8 T (CD45^hi^CD3^+^CD8^+^CD69^+^) cells, (**C**) cytotoxic CD8 T (CD45^hi^CD3^+^CD8^+^CD160^+^) cells, (**D**–**F**) CD8^+^IFN-γ^+^ T cells, (**G**) CD8^+^CD69^+^granzyme B^+^ T cells, (**H**) virus-specific CD8 T (CD8^+^gB tetramer^+^) cells, and (**I**) activated virus-specific CD8 T (CD8^+^gB tetramer^+^CD69^+^) cells by flow cytometry. (**J**) Primary mouse brain neurons (1 × 10^5^ cells) were infected with HSV-1 (MOI = 0.001) for 24 h, co-cultured without (–) or with 5 × 10^5^ CD8 T cells from brains of infected mice fed with control chow or the chow containing PLX5622 (PLX5622) and harvested 48 h later for virus titration. The mean of the neuron group was set as 100%. The data represent means + SEM (error bars) of 4–14 samples per group obtained from at least two independent experiments. *, *p* < 0.05.

**Figure 6 ijms-22-12457-f006:**
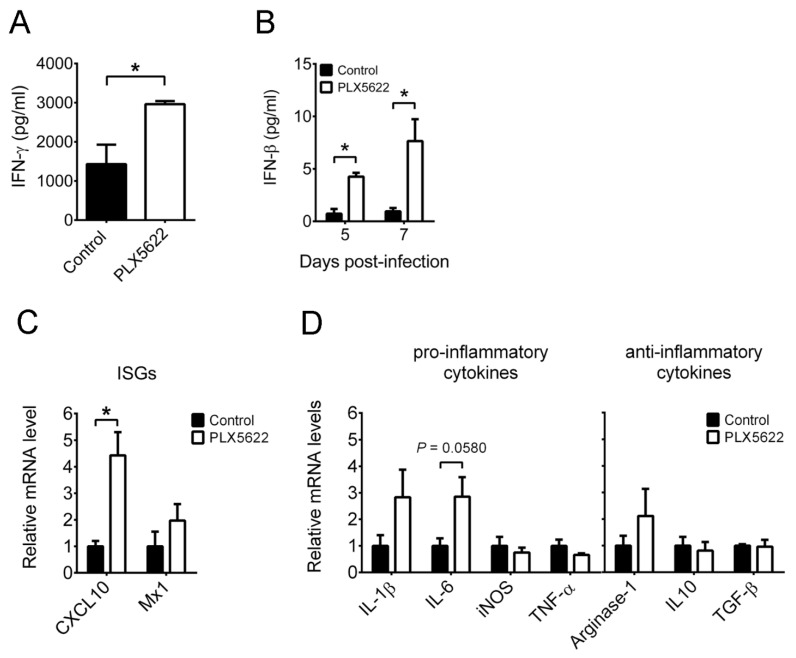
The effect of PLX5622 on the expression of immune modulators in brains of infected mice. Brains of infected mice fed with control chow or the chow containing PLX5622 were harvested, processed, and assayed by ELISA (**A**,**B**) and quantitative RT-PCR (**C**,**D**). The levels of (**A**) IFN-γ on day 7 post-infection and (**B**) IFN-β on the indicated days post-infection are shown. The mRNA levels of (**C**) IFN-stimulated genes (ISGs) and (**D**) pro-inflammatory as well as anti-inflammatory modulators on day 5 post-infection are shown. In panels (**C**,**D**), the data are expressed as the ratio of indicated mRNA normalized to β-actin mRNA, and the means of control groups were set as 1. The data represent means + SEM (error bars) of 4–5 samples per group. *, *p* < 0.05.

**Figure 7 ijms-22-12457-f007:**
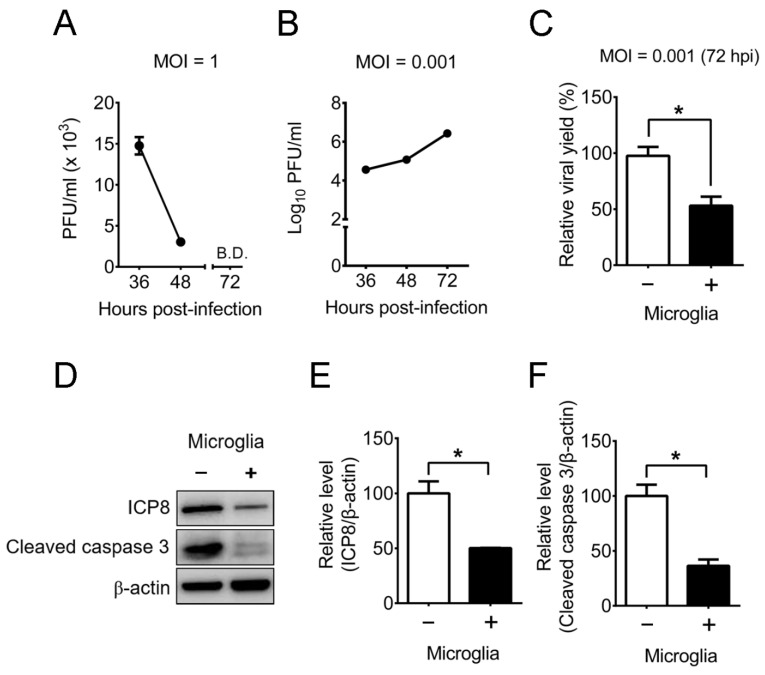
Mouse primary brain microglia reduce HSV-1 replication and apoptosis of mouse primary brain neurons. (**A**) Microglia and (**B**) neurons (1 × 10^5^ cells) cultured from mouse brains were infected with HSV-1 at the indicated MOIs and harvested for virus titration. B.D., below the detection limit. (**C**) Neurons (1 × 10^5^ cells) were co-cultured without (–) or with (+) 1 × 10^5^ microglia, infected with HSV-1 (MOI = 0.001), and harvested 72 h post-infection for virus titration. (**D**–**F**) Samples prepared as described in (**C**) were assayed for HSV-1 protein (ICP8), cleaved caspase 3, and β-actin by Western blotting. The levels of ICP8 or cleaved caspase 3 normalized to β-actin are shown. The means of neuron groups were set as 100% (**C**,**E**,**F**). The data represent means ± or + SEM (error bars) of 4–5 samples per data point and 3–16 samples per group obtained from at least two independent experiments. *, *p* < 0.05.

**Figure 8 ijms-22-12457-f008:**
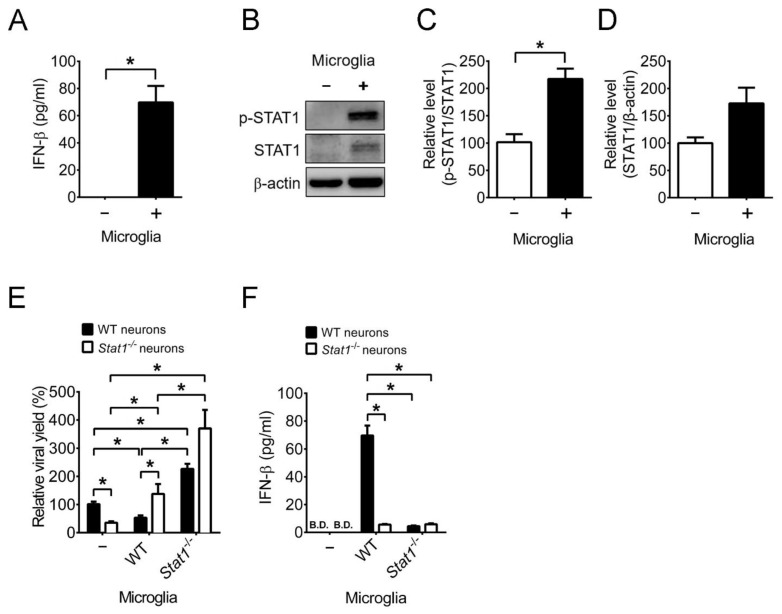
Microglia induce IFN-β and STAT1 phosphorylation to reduce the HSV-1 production in brain neurons. (**A**) Neurons (1 × 10^5^ cells) were co-cultured without (−) or with (+) 1 × 10^5^ microglia, infected with HSV-1 (MOI = 0.001), harvested 72 h post-infection, and assayed for IFN-β by ELISA. (**B**–**D**) Samples prepared as described in (**A**) were assayed for STAT1 by Western blotting. The relative levels of (**C**) Tyr701 phospho-STAT1 (p-STAT1)/STAT1 and (**D**) STAT1 normalized to β-actin are shown. Neurons (1 × 10^5^ cells) from wild-type (WT) or *Stat1^−/−^* mice were co-cultured without (–) or with 1 × 10^5^ microglia from WT or *Stat1^−/−^* mice, infected, and harvested as described in (**A**) for assays of (**E**) virus and (**F**) IFN-β. B.D., below the detection limit. The means of groups with WT neurons only were set as 100% (**C**–**E**). The data represent means + SEM (error bars) of 3–16 samples per group obtained from at least two independent experiments. *, *p* < 0.05.

## Data Availability

The data presented in this study can be found online. Further inquiries can be directed to the corresponding authors.

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
