# Peer review of "Microglia Reduce Herpes Simplex Virus 1 Lethality of Mice with Decreased T Cell and Interferon Responses in Brains"

_ijms, 2021, doi:10.3390/ijms222212457_

Round 1

Reviewer 1 Report

This manuscript by Tsai and colleagues deals with  mechanisms underlining the role of microglia in protecting mice from HSV-1 lethality. Although the manuscript is interesting and complex in its experimental design, as it reads results too confusing and difficult to understand in all its aspects. Furthermore it suffers of some issues that need to be fully addressed before the study could become suitable for publication in International Journal of Molecular Sciences, as detailed below.

Major issues

The first concern of this referee is related to the choice of the Authors to adopt the C57BL/6J mice. It is known that different mice strains react in completely different way to HSV-1 infection, certain mice being very refractory to viral infection/replication even in the brain. Furthermore, age and sex have an impact as well. The Authors need to discuss this aspect and to clearly state the rational of their choice, as this is essential for the relevance of the entire study. It is important to explain to Readers not completely familiar with this field why mice that present HSV-1 in the brain do not suffer ["All infected mice survived, and virus was detected in mouse brains on both days 5 and 7 post-infection (p.i.) (Figure 1A)]. Is this expected? It depends on the MOI adopted? It depends on the selected mice? Have then these results any impact on mechanisms protecting humans from fatal HSV-1 encephalitis?

It is not clear to this referee how many mice have been adopted for each experiment. The information is given only for some of the reported tests. However, it is a fundamental aspect. In line 170 the Authors state "all three mice...." thus I assume that for this experiment 3 mice were adopted for each condition. I am wondering if this number is sufficient to draw significant conclusions. The Authors should clearly state how many mice were used in each experiment and convincingly explain why they selected this number and if it is sufficient to support their conclusions.

While on one hand, I appreciated the number of different techniques the Authors adopted, I found a bit confusing the fact that different methods have been employed for addressing similar questions. As an example, why to adopt ELISA  to evaluate the expression of certain immune modulators in mice brain and Real Time PCR for additional ones. The Authors should adopt the same technique, possibly ELISA, for all the analyzed factors.

Lines 123-132: the experiment described here is not clear to me. First I do not exactly understand how it was performed neither reading the text, neither reading the Figure legend. Second, I do not understand how the microglia could have affected HSV-1 infectivity, as if I understood correctly also looking at the co-culture experiments, microglia is not infected by HSV-1. Do microglia produces factors that affect infectivity of HSV-1 which remains in the cell supernatant? This experiment should be clarified. Indeed, I am confused by the conclusion drawn by the Authors that they findings are consistent with "previous report of of robust expression of antiviral mediators in both mouse and human primary microglia during nonproductive HSV-1 infection". Then, microglia can be infected in a nonproductive manner? If this is correct, what do the Authors measured? Intracellular virus extracted from microglia? Please clarify

I am confused by the sentence reported at lines 158-159 "In control mice, virus was detected only in the brain, trigeminal ganglion, and eye, in which microglia are present [5, 31, 32]". Please clarify as I do not understand how this finding fits in the hypothesis of the Authors that microglia reduces viral antigens expression and viral mediated damage in mice brain. In this same experiment, the Authors should better explain how they harvested eyeballs, trigeminal ganglia, and whole brains for virus titration. A mention to a previous publication in the Materials and Methods section [55] is not enough. Can the Authors explain why the viral titer decreases 7 days p.i. in microglia depleted mice only in the trigeminal ganglion?

Figure 8B: I do not see STAT-1 in  the - lane, but only in the + one. How many times was this experiment repeated? It is not clear from the legend to the Figure (3-16 samples?). And if it was repeated as I guess from the quantification graphs, please show a gel were the un phosphorylated STAT1 is visible in the control.

The discussion of the discrepancy between this study and the results described in paper #23 is neither sufficient nor clear enough. This aspect is crucial and should be convincing.

In general is difficult to appreciate how many times the experiments were repeated and how many samples from independent experiments are reported in each Figure/Graph

Minor issues

Reviewer 2 Report

Dear authors,

This paper uses a murine infection model to analyse the interferon responses in microglia and the mechanisms that microglia protect from HSV-1 infection. The manuscript is well written and structured; however some major changes are required:

Line 25: “in vitro” in italic.

Line 32 (Introduction): In the introduction, the authors should explain what are the mechanisms of action of the interleukins and interferons that have been chosen for their study and why they have chosen them.

Line 143 (figures): In all figures, the authors indicate different p-values as significant. However, the statistical tests have been carried out with p-value < 0.05 as significant. Since a p-value less than 0.05 does not indicate greater significance of the results, or I recommend that the authors indicate only if the results present a p-value <0.05 or not.

Line 569: Many works have been published in relation to the use of different housekeeping depending on the tissue and the species. Authors should specify why they have chosen this housekeeping and not another. If no article confirms the stability of beta-actin in the analyzed tissue, the authors should normalize with several housekeeping to ensure the results obtained.

Line 599 (statistical analyses): The authors should explain the parametric and non-parametric test selected. Have they carried out a test of normality and homoscedasticity of the data previously? Specify it.

Round 2

Reviewer 1 Report

The Authors have addressed most of the points raised by this Referee. Thus the manuscript is now suitable for publication.

Reviewer 2 Report

Done.